# Increasing salinity of fibrinogen solvent generates stable fibrin hydrogels for cell delivery or tissue engineering

**Dillon K. Jarrell**[1], **Ethan J. Vanderslice**[1], **Mallory L. Lennon**[1], **Anne C. Lyons**[1], **Mitchell C. VeDepo**[1], **Jeffrey G. Jacot**[1,2]*

**1** Department of Bioengineering, University of Colorado Anschutz Medical Campus, Aurora, Colorado, United States of America, **2** Department of Pediatrics, Children's Hospital Colorado, Aurora, Colorado, United States of America

* jeffreyjacot@cuanschutz.edu

**Data Availability Statement:** All relevant data are within the paper and its Supporting information files; specifically, find raw data and analysis in the attached Excel file titled "Raw Data Statement."

## Abstract

Fibrin has been used clinically for wound coverings, surgical glues, and cell delivery because of its affordability, cytocompatibility, and ability to modulate angiogenesis and inflammation. However, its rapid degradation rate has limited its usefulness as a scaffold for 3D cell culture and tissue engineering. Previous studies have sought to slow the degradation rate of fibrin with the addition of proteolysis inhibitors or synthetic crosslinkers that require multiple functionalization or polymerization steps. These strategies are difficult to implement *in vivo* and introduce increased complexity, both of which hinder the use of fibrin in research and medicine. Previously, we demonstrated that additional crosslinking of fibrin gels using bifunctionalized poly(ethylene glycol)-n-hydroxysuccinimide (PEG-NHS) slows the degradation rate of fibrin. In this study, we aimed to further improve the longevity of these PEG-fibrin gels such that they could be used for tissue engineering in vitro or in situ without the need for proteolysis inhibitors. It is well documented that increasing the salinity of fibrin precursor solutions affects the resulting gel morphology. Here, we investigated whether this altered morphology influences the fibrin degradation rate. Increasing the final sodium chloride (NaCl) concentration from 145 mM (physiologic level) to 250 mM resulted in fine, transparent high-salt (HS) fibrin gels that degrade 2–3 times slower than coarse, opaque physiologic-salt (PS) fibrin gels both *in vitro* (when treated with proteases and when seeded with amniotic fluid stem cells) and *in vivo* (when injected subcutaneously into mice). Increased salt concentrations did not affect the viability of encapsulated cells, the ability of encapsulated endothelial cells to form rudimentary capillary networks, or the ability of the gels to maintain induced pluripotent stem cells. Finally, when implanted subcutaneously, PS gels degraded completely within one week while HS gels remained stable and maintained viability of seeded dermal fibroblasts. To our knowledge, this is the simplest method reported for the fabrication of fibrin gels with tunable degradation properties and will be useful for implementing fibrin gels in a wide range of research and clinical applications.

**Funding:** This work was supported by the National Science Foundation grant numbers GRFP 15385 (DKJ) and GRFP 181204 (MLL), the National Institute of Health grant numbers 5T32HL072738-16 (PI Dr. Robin Shandas supporting EJV) and 1R01HL130436 (JGJ), and the American Heart Association grant number 19POST34380541 (MCV).

**Competing interests:** The authors have declared that no competing interests exist.

## 1. Introduction

The development of thick tissues for the repair of large injuries or defects is a principal challenge for tissue engineering advancements [1–4]. *In vitro* tissue engineering approaches face the challenges of nutrient diffusion and recapitulation of tissue-specific signaling outside of the body while *in situ* tissue engineering approaches must promote normal tissue regeneration while avoiding fibrosis and aberrant tissue growth [5]. Both strategies must include scaffolds that support the proliferation and differentiation of tissue-specific stem cells while simultaneously degrading and being replaced with tissue-specific extracellular matrix [6,7].

Natural hydrogels make promising tissue engineering scaffolds because they are cytocompatible, bioactive, and readily remodeled by cells. The bioactivity of fibrin is particularly attractive since fibrin clots drive wound healing in the body by modulating inflammation, angiogenesis, and cell-matrix interactions [8–11]. Not surprisingly, fibrin glues and gels have been extensively investigated for use in wound sealing and the delivery of growth factors and cells. Despite this desirable bioactivity, the rapid degradation rate of fibrin has limited its usefulness as a scaffold for 3D cell culture and tissue engineering. Plasmin inhibitors such as 6-aminocaproic acid (ACA) prevent fibrinolysis but are difficult to implement *in vivo* [12]. Therefore, we aimed to develop a slowly-degrading ("stable") fibrin gel capable of supporting 1) the proliferation and differentiation of various stem cell types, 2) the development of a rudimentary capillary-like network *in vitro*, and 3) delivery and maintenance of cells *in vivo* without the need for degradation inhibitors.

Our group is interested in engineering tissues by differentiating amniotic fluid cells (AFC) or induced pluripotent stem cells (iPSC) in 3D. Most differentiation protocols for these two stem cell types require at least two weeks of culture, so a suitable fibrin scaffold must be stable when seeded with these cell types for at least two weeks. We previously reported that increased fibrin gel crosslinking using homobifunctional poly(ethylene glycol) n-hydroxysuccinimide (PEG-NHS) slows fibrin degradation when placed in media alone [13,14], but our follow-up work reported here demonstrates that even these PEG-fibrin gels degrade within one week when seeded with AFC (note that the PEG-fibrin gels reported in our previous studies are identical to the PS gels reported here, and are referred to here simply as fibrin gels because PEG is no longer an independent variable). To further reduce the rate of fibrin degradation, we sought to manipulate other fibrin composition variables. It is known that several properties of fibrin, including opacity, morphology, and mechanics, can be controlled by several composition variables, including pH, salinity, buffer type, crosslinkers, and the concentrations of fibrinogen, thrombin, and calcium [15–21]. Ferry et al. first reported that "fine" (transparent) and "coarse" (opaque) fibrin clots could be generated by adjusting pH and ionicity [22,23]; Eyrich and colleagues expanded on this work by demonstrating that only fine, transparent fibrin can maintain 3D chondrocyte culture without degrading [24]. A similar study by Davis and colleagues found that increasing sodium chloride in the gel precursor improved the gel mechanical properties and osteogenic behavior of seeded mesenchymal stem cells [25]. These results can likely be explained by the effects of chlorine ions on fibrin polymerization, as chlorine is known to bind fibrin and oppose lateral aggregation of protofibrils, resulting in thin, dense networks [26,27]. Another group decreased the rate of fibrinolysis by fusing an engineered peptide sequence derived from alpha-2 plasmin inhibitor to recombinant VEGF and covalently linking it to fibrin during gelation [28]. This strategy proved exceedingly effective, but is complicated, expensive, time consuming, and requires specific laboratory expertise. Robinson and colleagues also effectively decreased fibrinolysis using genipin crosslinking, which also exhibits neuritogenic effects [29]. While these effects were beneficial in their elegant formulation, they would be detrimental in most systems.

In this study, we found that increasing the salinity (NaCl concentration) of the fibrinogen precursor solution generates increasingly transparent fibrin with decreasing rates of degradation. We investigated the ability of these gels to support encapsulated induced pluripotent stem cells (iPSC) and amniotic fluid cells (AFC) *in vitro*. We also assessed the ability of the gels to support vascularization both *in vitro* and *in vivo*. Increased salinity during gelation did not affect the viability of encapsulated cells and the high-salt (HS) gels supported encapsulated AFC without degrading for at least two weeks. Both the physiologic-salt (PS) and HS gel formulations were able to maintain iPSC and support capillary-like network formation *in vitro* when seeded with human endothelial cells and fibroblasts. Finally, we found that HS fibrin gels are stable and maintain the viability of seeded cells when implanted subcutaneously into mice, while PS gels degraded completely within one week.

## 2. Materials and methods

### 2.1. Fibrin gel fabrication

Fibrin gels were fabricated in four steps. First, sterile fibrinogen from human plasma (Millipore Sigma, 341576) was dissolved in sterile PBS (Corning, 21-040-CM) at 80 mg/mL at 37°C for two hours. Second, PEG-NHS (3.4 kDa, SUNBRIGHT DE-034HS, NOF America Corporation) was dissolved in PBS at 8 mg/mL, syringe filtered, and immediately mixed with the dissolved fibrinogen 1:1 by volume (PEG:fibrinogen mole ratio of 10:1). The fibrinogen and PEG were allowed to crosslink for one hour at 37°C. Third, cells were dissociated with Accutase®, counted, and resuspended in growth media at 4X the desired final cell concentration. The 4X cell and fibrinogen-PEG solutions were mixed 1:1 by volume (PBS control added for cell-free gels) and the mixture was added to the appropriate culture vessel. Fourth, thrombin from human plasma (Millipore Sigma, 605190) was resuspended in cold calcium chloride solution (11.1 mM $CaCl_2$, 145 mM NaCl, pH 7.4 in DI water) at 20 U/mL. Thrombin solution was added to the cell-fibrinogen-PEG solution 1:1 by volume and quickly mixed by pipetting five times. Gelation occurred at 37°C for 5 minutes before gels were immersed in media or PBS. For cell experiments, media was replenished daily. PS and HS gels were fabricated with final concentrations of 10 mg/mL fibrinogen, 1 mg/mL PEG, 1 U/mL thrombin, 5 mM $CaCl_2$, and pH 7.4. PS fibrin was fabricated with a final NaCl concentration of 145 mM (physiologic concentration, total chlorine concentration of 155 mM), and HS fibrin was fabricated with a final NaCl concentration of 250 mM (total chlorine concentration of 260 mM). NaCl concentration was adjusted by adding NaCl to the PBS used as the solvent for the fibrinogen and PEG.

### 2.2. Measuring fibrin opacity

To assess which formulation variables are necessary to generate fine transparent fibrin, 50 uL gels were fabricated in the wells of a 96-well plate. Fibrin opacity was quantified by absorbance spectrophotometry at 352 nm (BioTek Synergy 2, Gen5 software). The gel formulation variables included final fibrinogen concentration (2.5, 5, 10, and 20 mg/mL), gel pH (6, 7, 7.4, 8, 8.5), final $CaCl_2$ concentration (0.1, 0.5, 5, 10, 25 mM), and final NaCl concentration (145, 175, 200, 250, 300 mM). These parameters were investigated independently; the constant gel formulation values were 10 mg/mL fibrinogen, pH 7.4, 5 mM $CaCl_2$, and 145 mM NaCl. Gel pH was adjusted by changing the pH in the $CaCl_2$ solution via 0.1 M NaOH. Salinity was adjusting by changing the NaCl concentrations in the PBS used for the fibrinogen and PEG solutions. The PEG:fibrinogen mole ratio remained 10:1 for all fibrinogen concentrations.

## 2.3. Scanning electron microscopy

Analysis of gel morphology was assessed using 200 uL fibrin gels in a 48-well tissue culture plate (Corning). After gelation, gels were hydrated in PBS for one hour, then dehydrated in ethanol (50%, 75%, 90%, and 100% ethanol for one hour each). Samples were attached to SEM stubs using double-sided carbon tape and coated with Au/Pd for 30 seconds using an EM ACE200 sputtercoater (Leica, Buffalo Grove, IL, USA). Images were collected with a JSM-6010LA SEM (JEOL, Tokyo, Japan). Analysis of fibers and pores in each gel were quantified using DiameterJ [30]. Each greyscale SEM image was segmented using the DiameterJ traditional segmentation algorithm to produce eight 8-bit black and white segmentations per SEM image. The best segmentation of the eight was selected based on the following criteria: (1) no partial fiber segmentations, (2) the intersections of fibers do not contain black spots (i.e. holes), (3) segmented fibers are representative of actual fibers in the image and are not background/imaging artifacts, and (4) segmentations accurately represent fibers' actual diameter. The selected segmentations were scaled and processed in DiameterJ to quantify pore and fiber characteristics.

## 2.4. Atomic force microscopy

To analyze compressive modulus at a micro scale, 30 uL fibrin gels were fabricated as drops in 35-mm dishes and submerged in PBS. AFM indentation experiments were performed with a NanoWizard 4a (JPK Instruments) using a cantilever with a nominal spring constant of 0.03 N/m and a pyramidal tip (MLCT-D, Bruker AFM Probes). The dishes were maintained at 37˚C during force measurements. Force curves were recorded from three separate regions on the surface of each hydrogel. At each region, 36 force curves were measured across a $100um^2$ area for a total of 108 measurements per gel. The data was analyzed using JPK image processing software. Young's modulus was calculated using the Hertz model of fit on the extend curve, and data is represented as mean +/- standard deviation.

## 2.5. Parallel plate rheology

To analyze the bulk Storage and Young's moduli, 160 uL HS and PS fibrin gels were fabricated as drops in 10-cm dishes and submerged in PBS at 37˚C overnight. Gels were carefully dislodged from the plate bottom and moduli were measured using a parallel-plate rheometer (Discovery Hybrid 2; TA Instruments) for five replicates of each hydrogel formulation. Samples were subjected to shear at 1% strain through a dynamic angular frequency range of 0.1 to 100 rad/s. Elastic modulus (E) was calculated from the linear region of the storage modulus (G') in the angular frequency sweep by assuming a Poisson's ratio of 0.5.

## 2.6. Gelation time

To analyze the gelation times of the HS and PS fibrin gels, 50 uL gels were fabricated on a parallel-plate rheometer (Discovery Hybrid 2; TA Instruments). 25 uL of fibrinogen solution was added as a drop to the 37˚C base plate, then the top plate was lowered to contact the drop before being raised to a gap distance of 800 um. Measurement of the storage modulus was started using 1% strain and 10 rad/s angular frequency. After 20 seconds, 25 uL of thrombin solution was added to the fibrinogen solution and mixed by repeat pipetting 3 times. Measurement was allowed to continue until the modulus plateaued (approximately 6 minutes). This method was not used to calculate gel Young's modulus because the angular frequency chosen here was chosen arbitrarily because it effectively detected stiffening during gelation. Any

change of the angular frequency for the measurement of gelation time drastically changed the measured initial and final values of storage moduli.

## 2.7. Swelling ratio

To assess swelling ratio, 50 uL fibrin gels were fabricated at 145, 175, and 250 mM NaCl as drops in a 10 cm tissue culture-treated dish (Corning). After equilibrating in sterile PBS at 37˚C overnight, the gels were scraped from the plate using a cell scraper and the wet weight was obtained after dabbing off excess PBS. Swelling ratio was calculated as the measured wet weight minus the hypothetical dry weight (50 uL of 10 mg/mL PEGylated fibrin) divided by the hypothetical dry weight. Gels were not dried and rehydrated.

## 2.8. Growth factor release

To assess the ability to release growth factors, 50 uL HS (250 mM NaCl) and PS (145 mM NaCl) gels (10 mg/mL fibrinogen) were fabricated with 100 ng/mL FGF-2 (Peprotech, 100-18b) and 100 ng/mL VEGF-165 (Shenandoah Biotechnology, 100–44) in the bottom of 1.6 mL Eppendorf tubes. To fabricate these gels, a PBS solution of 400 ng/mL FGF-2 and 400 ng/mL VEGF-165 was used instead of the normal 4X cell solution in Step 3 of Section 2.1 above. After gelation, 1 mL of 0.5% protease-free BSA (Sigma, A3059) in PBS was added to each tube and was completely replaced daily. After 0, 1, and 7 days, the supernatant was removed and 50 uL of 0.2 mg/mL human plasmin (Enzyme Research Laboratories) in HEPES buffer (pH 8.5, Boston Bioproducts) was added. Fibrin degradation occurred overnight at 37˚C. Growth factor retention was measured using ELISAs (Peprotech, 900-K08 and 900-K10) and a BioTek Synergy 2 luminescent plate reader (Gen5 software).

## 2.9. Papain-mediated degradation

To analyze the fibrin degradation kinetics as a function of salinity, 75 uL fibrin gels were fabricated at 145, 175, and 250 mM NaCl as drops in 10 cm tissue culture-treated dishes and were allowed to equilibrate in PBS at 37˚C for one hour. At time zero, 10 mL of warm protease solution consisting of 0.8 uM Papain (21 kDa, Sigma, P4762) and 2.7 mM N-acetylcysteine (Sigma, A7250) in PBS was added to each dish. Every 10 minutes for one hour, three gels from each formulation were scraped from the dish, dabbed on a KimWipe, and weighed. Since we were not concerned with preservation of encapsulated proteins or cells, papain was selected as a protease for this experiment over the more expensive, fibrin-specific protease plasmin.

## 2.10. Cell viability and cell-mediated degradation

To analyze cell viability in the HS fibrin gels, passage 3–5 AFC were dissociated and resuspended in EGM-2 (Lonza) at $4x10^5$ cells/mL. 75 uL HS (250 mM NaCl) and PS (145 mM NaCl) fibrin gels were fabricated using the 4X AFC suspension as drops in wells of a 6-well tissue culture-treated plate (three gels per well). After gelation, 2 mL of EGM-2 was added to each well and the gels were incubated at 37˚C and 5% $CO_2$. After 1, 24, and 96 hours, EGM-2 was aspirated and cells were stained using the fluorescent LIVE/DEAD Viability/Cytotoxicity Kit (Invitrogen) according to kit instructions. Three images were captured from each gel using a Zeiss Observer.Z1 and long-distance objective (LD Plan-NEOFLUAR 20X/0,4 Ph2) and were used to count living and dead cells.

To analyze the cell-mediated degradation kinetics of HS and PS fibrin formulations, gels were fabricated with a final concentration of $1x10^5$ AFC/mL (P3-5) and incubated in EGM-2 +/- 1 mg/mL of the plasmin inhibitor 6-aminocaproic acid (Sigma, A2504) at 37˚C and 5%

$CO_2$. After 0, 7, and 14 days, gels were imaged using phase contrast (Zeiss ObserverZ.1) and wet weights were recorded.

## 2.11. iPSC encapsulation

iC4-4 iPSC [31] were purchased from the Gates Center for Regenerative Medicine at the University of Colorado. iPSCs were maintained in mTeSR-1 (StemCell Technologies) on 6-well tissue culture plates coated with Matrigel® (Corning). Media was replenished daily, and cells were passaged as colonies every 4–5 days using 0.5 mM EDTA. iPSC (P20-30) were dissociated with Accutase®, centrifuged for 3 minutes at 300 rcf, and resuspended in mTeSR-1 with 10 uM ROCK inhibitor (Y-27632, Sigma) at $4x10^6$ cells/mL. 30 uL HS and PS fibrin gels were fabricated using the 4X iPSC suspension as drops in a 6-well plate (three gels per well). After gelation, 2 mL of mTeSR-1 plus 10 uM ROCK inhibitor was added, after which mTeSR-1 alone was replenished daily. After 24 and 96 hours, gels were imaged using phase contrast and analyzed for pluripotency markers using PCR. Briefly, to analyze mRNA expression, whole gels were scraped from the plate and homogenized in TriZol (Life technologies). RNA was extracted in chloroform and washed using the Qiagen RNeasy Minikit. Reverse transcription was conducted using the High-Capacity cDNA Reverse Transcription Kit (Applied Biosystems) according to kit instructions. Pluripotency genes POU5F1 (ThermoFisher, hs04260367_gH) and NANOG (ThermoFisher, hs04260366_g1) were measured, and Ct values were normalized to GAPDH (Thermofisher, hs02758991_g1) and gene expression of monolayer iPSC cultured on Matrigel® at matching timepoints ($2^{\Delta\Delta Ct}$ method).

## 2.12. Capillary-like network formation *in vitro*

P3-5 GFP-HUVECs (Angio-Proteomie cAP-0001GFP) and P3-5 human dermal fibroblasts (HDFs, Lonza CC-2509) were dissociated, counted, and resuspended in EGM-2 at $3.2E^6$ cells/mL (4:1 HUVEC:HDF ratio). This 4X cell solution was used to fabricate 100 uL HS and PS gels in wells of a 48-well tissue culture plate. After 7 days, gels were analyzed for network formation using immunofluorescence. Briefly, whole gels were fixed in 4% paraformaldehyde for 30 minutes at room temperature, washed, blocked with 3% BSA and 1% FBS in PBS for one hour, and stained with anti-α-smooth muscle actin (Sigma, C6198, 1:200) in blocking solution for two hours. After three 15-minute PBS washes, cells were stained with DAPI and imaged using a Zeiss ObserverZ.1 fluorescent microscope.

## 2.13. Subcutaneous injection of fibrin gels

*In vivo* degradation of fibrin gels was assessed through subcutaneous injection of cell-seeded HS and PS gels in athymic nude mice (6–7 weeks old, Foxn1[nu]; Envigo) in a protocol approved by the Institutional Animal Care and Use Committees at the University of Colorado Anschutz Medical Campus (protocol #00564). Injections were performed as previously described [23]. Briefly, 500-uL HS and PS gels + $5E^5$ cells/mL were fabricated in 1-mL syringes (n = 4 gels per group). The cell suspension contained a 2:1 ratio of GFP-HUVECs (Lonza) and HDFs. Mice were anesthetized with isoflurane and two gels were implanted into opposite dorsal, posterior pockets. After 7 days, mice were sacrificed and fibrin hydrogels were explanted while retaining the surrounding tissue.

## 2.14. Histology

Tissues were fixed in formalin for 48h and sent to the Biorepository Core Facility at the University of Colorado Anschutz Medical Campus for paraffin embedding, sectioning, and H&E

staining. For immunofluorescence analysis, sections first underwent antigen retrieval via incubation in citrate buffer (10 mM citric acid, 0.05% tween, pH 6.0) at 95C for 10 minutes. To analyze cell delivery and morphology, slides were stained with anti-Vimentin (Sigma, C9080, 1:200), and DAPI.

### 2.15. Statistics

Data is presented as mean +/- standard error of mean, unless otherwise noted. One-way analysis of variance (ANOVA) followed by a *post hoc* bonferroni correction for multiple comparisons was performed for all comparisons. A value of $p < 0.05$ was considered significant in all tests.

## 3. Results and discussion

### 3.1. Fine, transparent fibrin gels can be fabricated by increasing NaCl concentration alone

Motivated by the findings of other groups that fibrin degradation rate is correlated to its opacity [22–24], we generated fibrin gels with varying pH and concentrations of thrombin, fibrinogen, $CaCl_2$, and NaCl and proceeded to measure gel opacity. In contrast to the study by Eyrich et al. [24], we found that fibrin opacity does not depend on formulation pH or $CaCl_2$ concentration (Fig 1A and 1B) and that fibrin opacity always increases as fibrinogen concentration increases (Fig 1C). Instead, we report the novel finding that simply increasing the concentration of NaCl in the fibrinogen solution yields transparent fibrin gels (Fig 1D and 1E). Since this trend plateaus above a final NaCl concentration of 250 mM (Fig 1D), this formulation (denoted HS fibrin) was selected for the remaining experiments.

The discrepancies between our observations and previous literature are difficult to explain. We used >90% clottable human fibrinogen, while the Eyrich study used only 60% clottable bovine fibrinogen [24]. Furthermore, the Eyrich study used aprotinin and Baxter dilution buffer, which may have changed their gel ionicity compared to our parameters. The published dependence of gel opacity on $CaCl_2$ concentration was of particular interest to us because we did observe that gel opacity depended on NaCl concentration, both of which yield chloride ions in solution. An additional literature search led us to hypothesize that the increased concentration of chloride ions was responsible for the change in gel opacity [26]. We fabricated gels using 250 mM NaCl (final concentration of 270 mM Cl⁻ when including $CaCl_2$ in the thrombin solution) and they were transparent. However, when we fabricated gels using 65 mM $CaCl_2$ rather than 20 mM $CaCl_2$ (total final concentration of 275 mM Cl⁻ when including NaCl in the fibrinogen solution), gels remained opaque. Therefore, it is difficult to maintain the chloride ions alone are able to cause the observed difference in gel opacity.

### 3.2. Transparent gels exhibit altered gel morphology and hydrogel properties

Having fabricated transparent fibrin gels, we next sought to analyze differences in gel morphology. SEM imaging confirmed that the turbid, coarse PS gels consist of a loose network of fibrils while the fine HS gels consist of a dense network of significantly thinner fibrils (Fig 2A–2C), consistent with previous studies [25,26]. Parallel plate rheology revealed that increased NaCl increased the bulk Storage and Young's moduli of the hydrogels (3.58 kPa for HS gels versus 0.57 kPa for PS gels, Fig 2E and 2F). AFM analysis revealed that the fibrils of the turbid PS gels were exceedingly varied in Young's modulus; values ranged from 0.5 kPa to 158 kPa and were clearly more heterogeneous than HS gels in SEM images (Fig 2D). In contrast, the

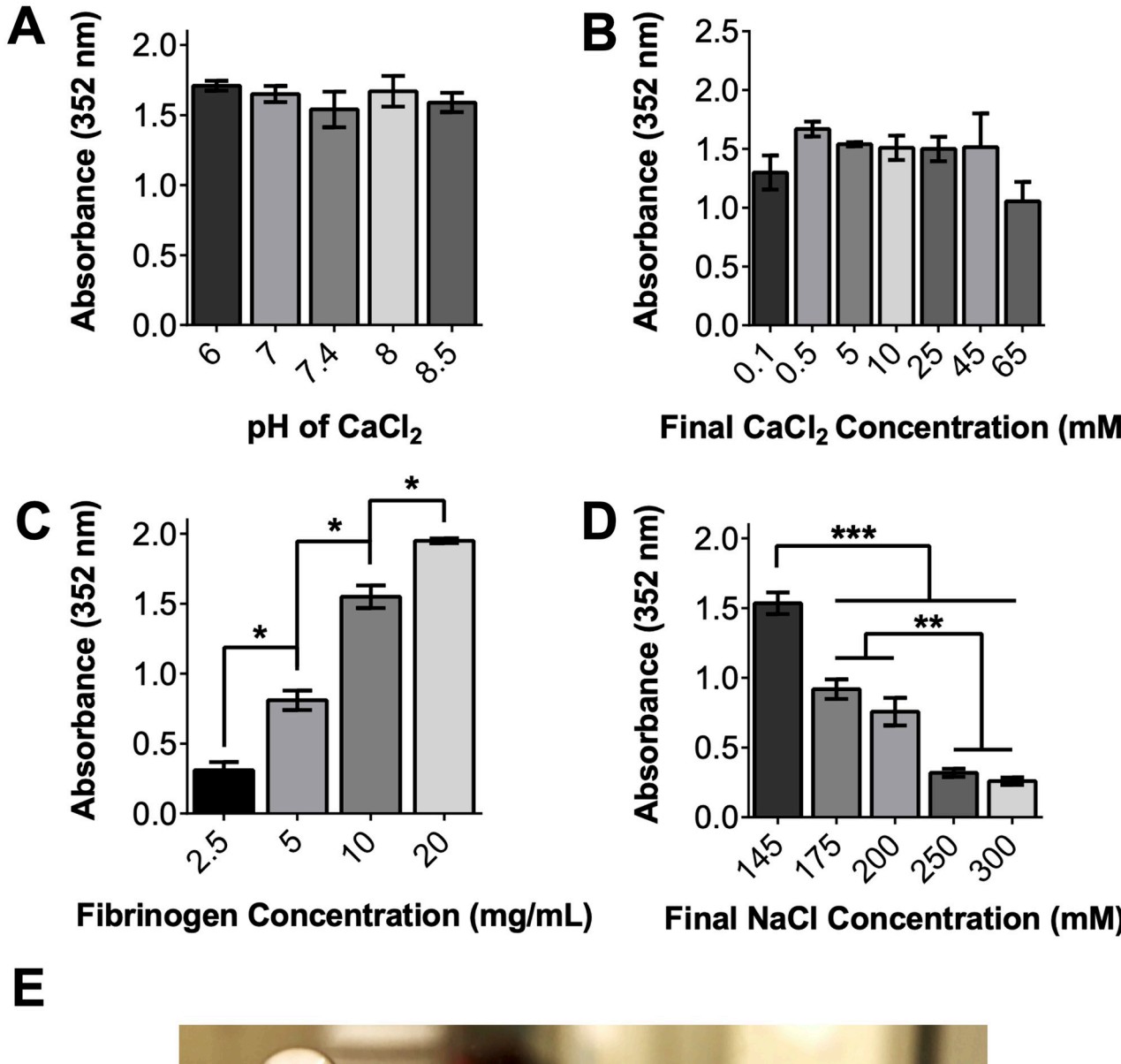

**Fig 1. Increasing salinity of fibrinogen solvent with NaCl impacts fibrin opacity.** pH (A) and CaCl₂ concentration (B) of gel precursor solutions do not impact gel transparency or morphology. Increasing fibrinogen concentration decreases transparency (C). However, increasing NaCl concentration in fibrinogen solution prior to gelation increases final gel transparency (D, E). For reference, PBS absorbance of 352 nm light is 0.06.

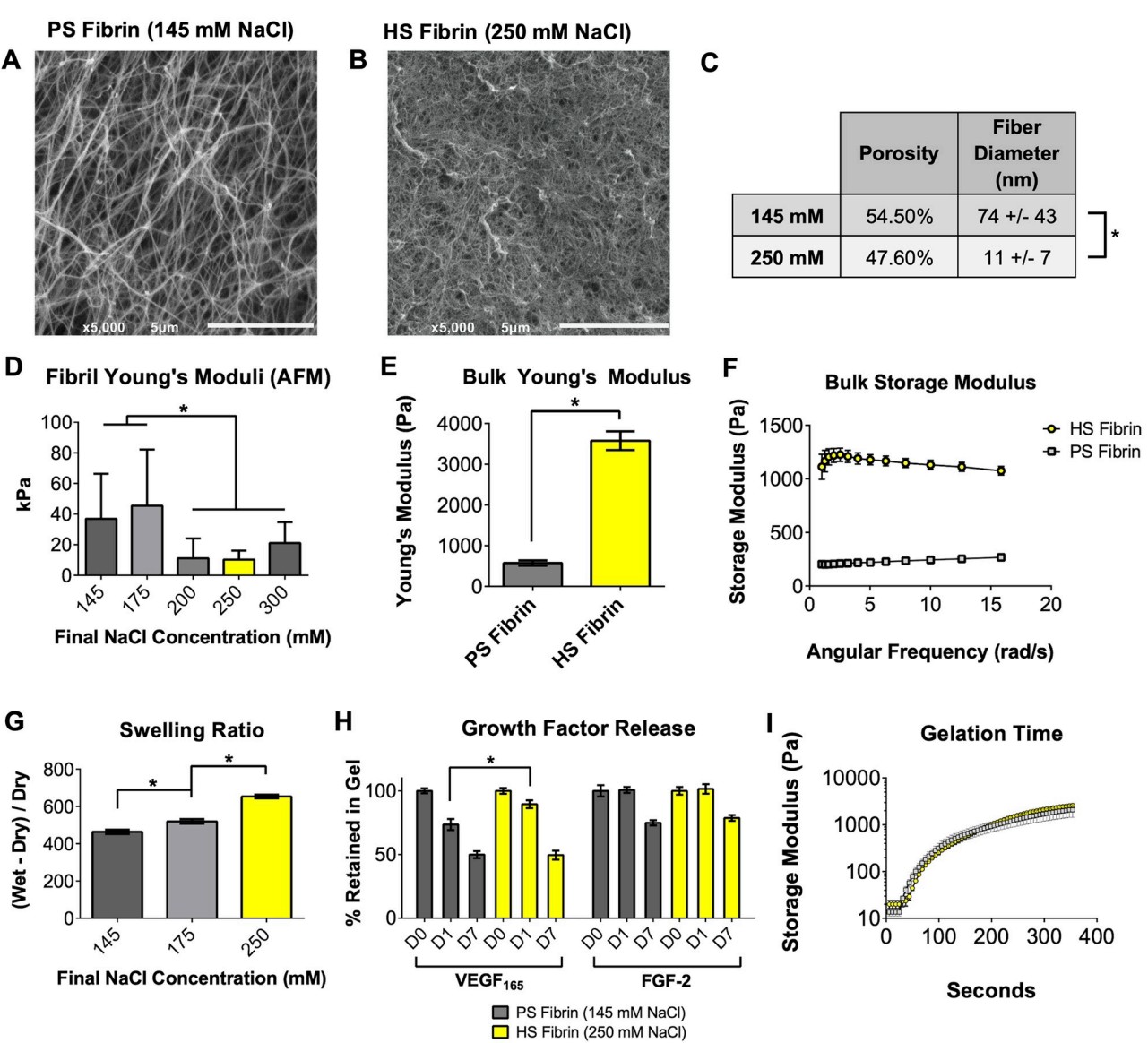

**Fig 2. Changing salinity of fibrinogen solution alters fibrin gel properties.** SEM images of fiber morphology (A, B) reveal that increasing salt concentration decreases porosity and fiber size (C). Young's moduli of individual fibers measured by AFM (D), Young's and Storage Modulus of the bulk materials measured by parallel plate rheometry (E,F), Swelling Ratio (G), and growth factor sequestering (F) are also significantly affected by salinity during gelation.

elastic moduli of the HS fibrils were more homogeneous at all NaCl concentrations greater than or equal to 200 mM, perhaps indicating a critical salt concentration that is able to alter the polymerization of 10 mg/mL fibrinogen. Analysis of HS and PS fibrin gelation times revealed that the gelation kinetics at 37°C were not significantly different (Fig 2F), suggesting that any ionic inhibition of fibril aggregation does not alter the rate at which thrombin cleaves fibrinogen or the rate at which the the gel forms, but rather alters the type of fibrlis being formed. In addition, HS gels were larger in volume than PS gels of equal fibrinogen content after reaching equilibrium (Fig 2G), implicating altered water-handling due to the differences in fiber morphology.

Finally, since fibrin is known to sequester growth factors through its heparin-binding domain [11], we hypothesized that the increased number of fibrils in HS gels would better sequester the angiogenic growth factors FGF-2 and VEGF$_{165}$. As expected, the HS gels released VEGF$_{165}$ more slowly than the PS gels, however both gel types released FGF-2 at the same rate and reached the same steady-state value of FGF-2 and VEGF$_{165}$ retention after 7 days (Fig 2H).

### 3.3. Transparent HS fibrin is stable and supports formation of capillary-like networks *in vitro*

Next, we examined how increased salt concentrations and the resulting altered fibrin morphologies influence encapsulated cell viability, gel degradation rate, and capillary-like network formation. Amniotic fluid cells (AFCs) were selected for testing because of their mesenchymal stem cell-like potency and their usefulness in pediatric regenerative medicine (S1 Fig) [32,33]. Since cells encapsulated in HS gels are briefly exposed to super-physiologic NaCl concentrations during fabrication, we examined cell viability after encapsulation using a LIVE/DEAD® immunofluorescence assay (Fig 3A–3F). Compared to cells in PS gels, we found no difference in AFC viability at early (1h, 24h) or late (96h) timepoints (Fig 3I).

We previously demonstrated that PS gels support capillary-like network formation *in vitro* [13,14]. To confirm that HS fibrin retains this ability, we co-seeded human dermal fibroblasts (HDF) and green fluorescent protein-labeled human umbilical vein endothelial cells (GFP-HUVEC). After 7 days, whole gels were fixed, stained with anti-α smooth muscle actin (α-SMA) and imaged. HS and PS gels both supported branching and network formation, with clear overlap of GFP-HUVEC and α-SMA-expressing HDF (Fig 3G and 3H). While we did not find any evidence of lumen formation, it does appear that the networks formed in HS fibrin are thinner than those formed in PS fibrin. In future work, HS fibrin may be studied as a scaffold capable of generating very small blood vessels, which remains a challenge in the field.

Finally, PS and HS gel degradation was investigated using a non-specific protease or encapsulated AFCs. When treated with papain (non-specific protease), the degradation rate decreased as salt concentration increased (Fig 4A). PS gels degraded approximately two times faster than HS gels and also more quickly than gels formed with an intermediate salt concentration (175 mM NaCl). This confirmed the observation made by Eyrich et al. that transparent fibrin gels are more stable [24], as well as confirmed our hypothesis that increased ionicity

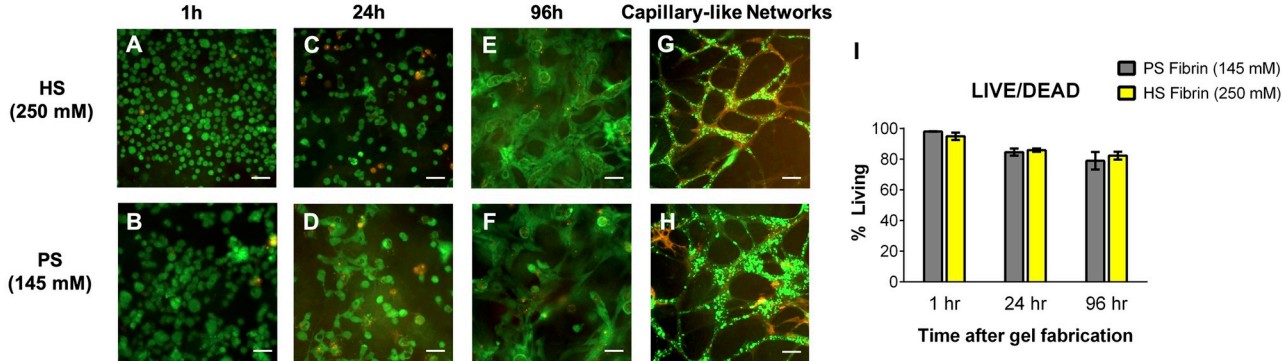

**Fig 3. Increased salinity does not affect viability of encapsulated cells or ability to form endothelial networks.** LIVE/DEAD® analysis stains Live cells green and Dead cells red. No significant difference in viability detected between amniotic fluid cells seeded in HS and PS gels (A-F), as quantified in (I). Both HS and PS gels support capillary-like network formation when seeded with GFP-HUVEC (green) and HDF (stained red with anti-α-smooth muscle actin) (G, H). Scale bars 50um.

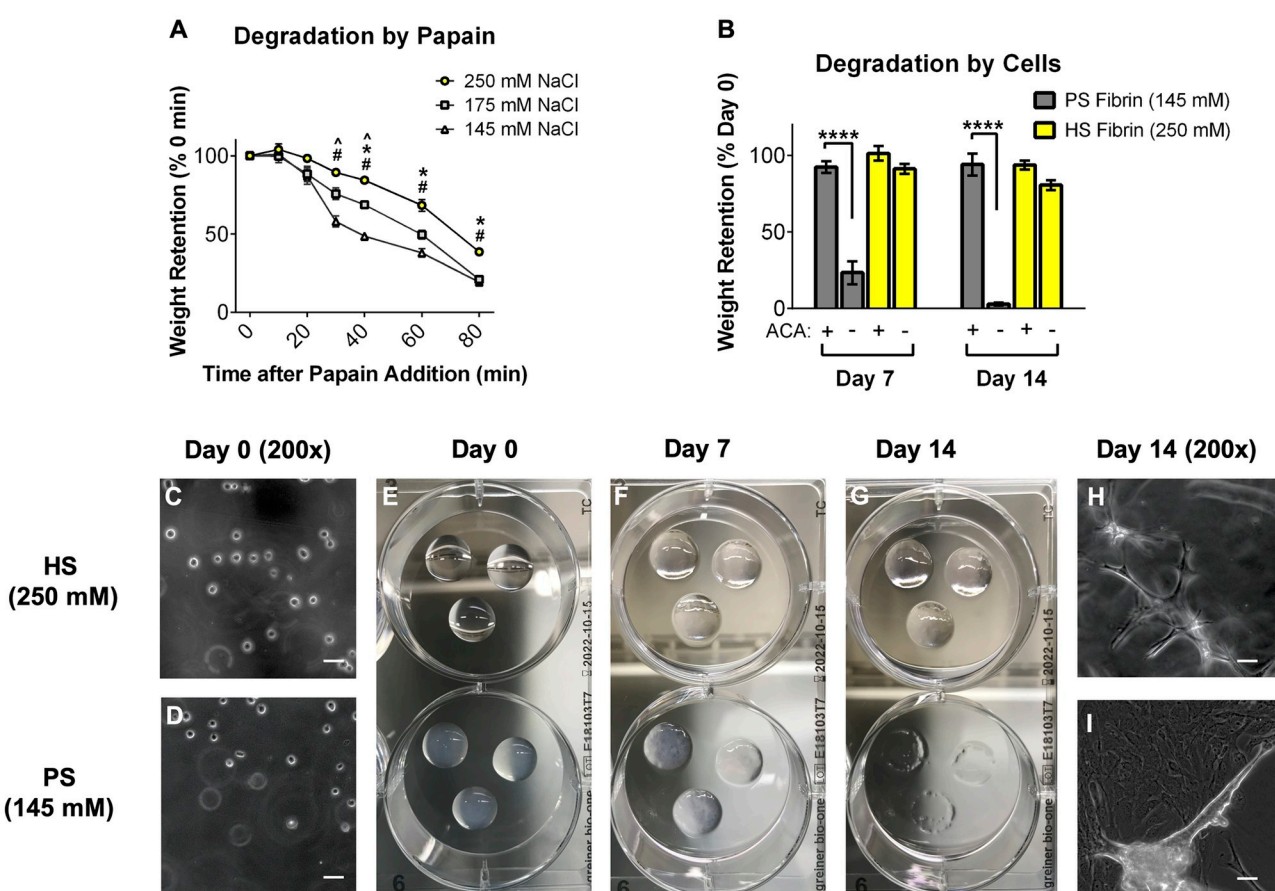

**Fig 4. Transparent high salt (HS) fibrin degrades slowly *in vitro*.** Increasing salt concentration yields fibrin with decreasing degradation kinetics when treated with Papain (no cells, panel A) and when seeded with AFC (B). 1 mg/mL 6-aminocaproic acid (ACA) prevents cell-mediated fibrinolysis, but HS gels are stable without ACA for at least 14d. Images of AFC-seeded gels degrading over 14d shown in panels C-I. By day 14, PS gels are completely degraded while HS gels are stable and continue to support 3D AFC culture (G-I). Scale bars of 200x images = 50um.

leads to a more stable gel. A similar trend was observed when gels were degraded by encapsulated AFC (Fig 3C and 3D). By day 7 post-encapsulation, PS gels were significantly degraded while HS gels remained intact (Fig 4B and 4F). After two weeks of culture, PS gels were nearly-completely degraded, leaving a few gel remnants and a monolayer of viable cells on the surface of the plate (Fig 4G–4I). In contrast, HS gels retained 80.6 +/- 7.8% of their wet weight and continued to support 3D culture of proliferating AFC (Fig 4B and 4G–4I). In both groups, fibrinolysis was prevented with the addition of 1 mg/mL ACA to EGM-2 (Fig 4B), which confirms that gel degradation is due to plasmin release from seeded cells. Interestingly, while the PS gels degraded significantly after 7 days when seeded with AFC only, the PS gels were stable after 7 days when seeded with HDF and HUVEC (Fig 3H). This is likely because HDF rapidly produce extracellular matrix proteins like collagen, which offset the concurrent degradation of fibrin.

## 3.4. Transparent HS fibrin is stable *in vivo* and maintains viability of delivered cells

After determining that HS gels are more stable *in vitro* when seeded with AFCs or treated with proteases, we sought to investigate the behavior of the gels *in vivo*. We have previously shown

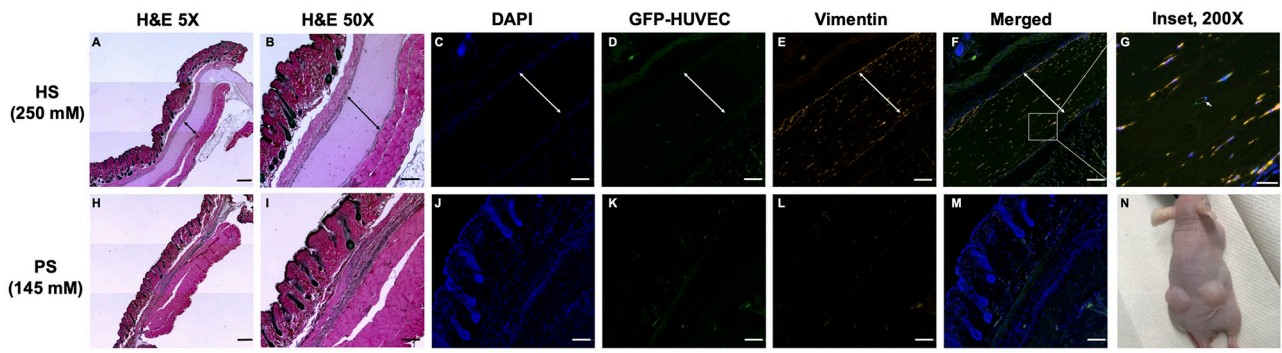

**Fig 5. HS gels are stable *in vivo* after 7d, but PS gels degraded completely.** HS and PS precursor solutions were mixed with GFP-HUVEC (green) and HDF (stained with Vimentin, red) and injected subcutaneously into athymic mice (N). After 7d, HS gels remained intact (H&E, A and B) and delivered cells remained viable (IF, C-G). Remaining gel indicated with arrows. PS gels degraded completely; no gel or delivered cells were detectable after 7d (H-M). Scale bars 500um (A, H), 200um (B-M), 50um (G).

that the inclusion of PEG in our PS fibrin gels improves longevity *in vivo* [13], and we followed the same subcutaneous gel injection protocol to assess differences between HS and PS hydrogel degradation and ability to support delivered cells. Furthermore, we included GFP-HUVEC and HDF to see if the gel(s) could support vascularization and angiogenesis once injected. HS and PS gels were seeded with GFP-HUVEC and HDF, injected subcutaneously into athymic nude mice, and explanted one week later (Fig 5N). Hematoxylin and eosin staining clearly revealed that the HS gels remained stable and intact after one week while the PS gels were completely degraded (Fig 5A, 5B, 5H and 5I). *In vitro*, the HDFs prevented even PS gel degradation. However, it is likely that the host innate immune system increased proteolysis *in vivo*, resulting in PS gel degradation despite HDF seeding. Further analysis using anti-Vimentin staining and fluorescent imagining revealed viable HDF throughout the stable HS gels interspersed with GFP-HUVEC, but little to no capillary formation (Fig 5C–5G). While α-SMA was used for *in vitro* staining, vimentin was used for the *in vivo* portion of this work because we found that it specifically labeled the delivered HDFs. In contrast, the explanted skin and underlying fat and muscle tissue surrounding the PS gels revealed no surviving HDF, no GFP-HUVEC, and no gel remnants (Fig 5J–5M). Initially, we hypothesized that the HS gels would be most useful for large tissue defect applications and that the PS gels would be most useful for rapid cell delivery applications, however this experiment suggests that the HS gels maintain cell viability better than the PS gels and may be superior for both applications. Surprisingly, we detected no angiogenesis into the injected HS fibrin. This could be due to the relatively short implant time (7 days) but is more likely due to the fact that no significant injury was caused by the subcutaneous injection of the gels. As a result, a healing response that would have driven angiogenesis was probably not triggered in the mouse. We would expect angiogenesis if the HS fibrin was used in the repair of a wound or heart defect.

### 3.5. HS and PS fibrin gels are both capable of iPSC maintenance in 3D

Recently, several studies have sought to identify materials capable of supporting iPSC for expansion and tissue engineering. Fibrin(ogen) has been found to be capable of this maintenance [34,35]. Because our group is interested in 3D stem cell differentiations with multiple stem cell types, we sought to corroborate these findings and determine if gel structure influences 3D iPSC culture. We encapsulated iPSC in PS (coarse) and HS (fine) gels and assessed pluripotent gene expression and qualitative cell morphology. Previous groups have shown that encapsulated iPSC must be allowed to proliferate for at least three days after seeding to lead to

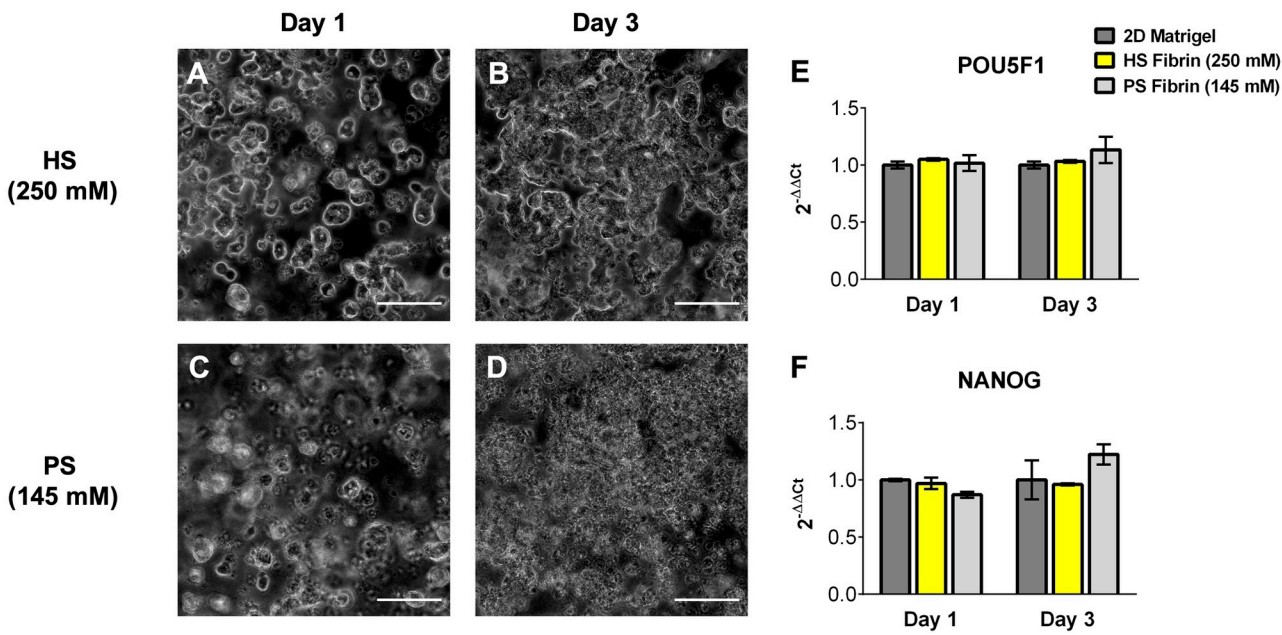

**Fig 6. PS and HS gels support iPSC expansion and pluripotency.** Both gel types maintain expression of pluripotency genes POU5F1 (E) and NANOG (F) compared to standard 2D Matrigel® culture. HS gels appear to drive formation of spheroid-like iPSC colonies (A,B) while PS gels appear to maintain singularized iPSC (C,D). Scale bars 50um.

successful differentiations, so we assessed pluripotent gene expression after 1 and 3 days [35]. After culturing iPSC within the two gel formulations and on Matrigel® (2D culture), we observed no difference in pluripotent gene expression (POU5F1 and NANOG) 1 and 3 days after passage among any of the culture groups (Fig 6E and 6F). Interestingly, iPSC in the fine HS gels formed large and compact 3D colonies, while iPSC in the looser PS gels appeared to remain mostly singularized, though this finding was purely qualitative and was not measured (Fig 6).

## 4. Conclusions

In the field of tissue engineering, there is a need for simple scaffolds that support angiogenesis, cell proliferation and remodeling, and exhibit tunable degradation rates. Fibrin gels are particularly promising because of their bioactivity; fibrin modulates the healing response *in situ* during wound healing. However, the rapid degradation rate of fibrin has limited its usefulness in 3D cell culture and tissue engineering. In this work, we demonstrated that the fiber architecture and degradation rate of PEGylated fibrin can be tuned simply by changing the NaCl concentration in the fibrinogen solvent. Using our simple four-step fabrication method, increasing the NaCl concentration in the fibrinogen solvent from 145 mM to 565 mM results in HS PEG-fibrin gels with a final NaCl concentration 250 mM. These HS gels exhibit fine fiber morphology, rendering them transparent compared to the opaque, coarse-fiber PS gels. The increased transparency of HS gels could be useful for 3D imaging experiments, but perhaps even more useful is our finding that HS gels degrade approximately three times more slowly than PS gels without affecting seeded cell viability, capillary-like network formation, or maintenance of iPSC. Furthermore, HS fibrin gels are stable *in vivo* and maintain the viability of delivered cells better than PS gels. Our previous work demonstrated that PEGylation of the fibrinogen also lowers the degradation rate of fibrin hydrogels, and for this reason we used

only PEGylated fibrin gels. However, we expect that even without PEGylation, HS fibrin gels would exhibit superior transparency and degradation kinetics versus PS gels. To our knowledge, this work represents the simplest reported method for controlling the transparency and degradation rate of fibrin without the need for fibrinolysis inhibitors. In our future studies, we will differentiate AFC and iPSC within the stable HS fibrin to create useful tissues for therapeutic implant, disease modeling, and drug screening. Specifically, we are interested in differentiating cardiac tissues and assessing this stable fibrin formulation in the repair of structural heart defects. Other future work should investigate the utility of this stable, transparent fibrin in other clinical and tissue engineering applications including wound healing (perhaps using a murine diabetic wound model), surgical glues (perhaps using AFM adhesion testing), cell delivery, developmental studies, and 3D cell culture and imaging.

## Supporting information

**S1 Fig.**
(TIF)

**S1 File.**
(DOCX)

**S1 Data. Data statement.**
(XLSX)

## Acknowledgments

The authors would like to acknowledge Dr. Eric Wartchow and the Colorado Children's Hospital Microscopy Core for assistance with SEM imaging, the University of Colorado Flow Cytometry Core for assistance with amniotic fluid cell FACS, Emily C. Beck, PhD, for her scientific expertise, Duncan Davis-Hall and the Chelsea Magin laboratory for assistance with bulk rheometry, and the University of Colorado Animal Care Facilities for assistance in animal handling training and care.

## Author Contributions

**Conceptualization:** Dillon K. Jarrell, Ethan J. Vanderslice, Jeffrey G. Jacot.

**Data curation:** Dillon K. Jarrell, Anne C. Lyons.

**Formal analysis:** Dillon K. Jarrell, Mallory L. Lennon, Anne C. Lyons.

**Funding acquisition:** Dillon K. Jarrell, Jeffrey G. Jacot.

**Investigation:** Dillon K. Jarrell, Ethan J. Vanderslice, Mallory L. Lennon.

**Methodology:** Dillon K. Jarrell, Jeffrey G. Jacot.

**Software:** Anne C. Lyons.

**Visualization:** Mitchell C. VeDepo.

**Writing – original draft:** Dillon K. Jarrell.

**Writing – review & editing:** Dillon K. Jarrell, Ethan J. Vanderslice, Mallory L. Lennon, Mitchell C. VeDepo, Jeffrey G. Jacot.

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
