## [Decision Letter · Decision Letter 0]

28 Sep 2020

PONE-D-20-26975

Increasing salinity of fibrinogen solvent generates stable fibrin hydrogels for cell delivery or tissue engineering

PLOS ONE

Dear Dr. Jacot,

Thank you for submitting your manuscript to PLOS ONE. After careful consideration, we feel that it has merit but does not fully meet PLOS ONE’s publication criteria as it currently stands. Therefore, we invite you to submit a revised version of the manuscript that addresses the points raised during the review process.

We look forward to receiving your revised manuscript.

Kind regards,

Esmaiel Jabbari, PhD

Academic Editor

PLOS ONE

Journal Requirements:

Reviewers' comments:

Reviewer's Responses to Questions

**Comments to the Author**

1. Is the manuscript technically sound, and do the data support the conclusions?

Reviewer #1: Yes

Reviewer #2: Yes

Reviewer #3: Partly

2. Has the statistical analysis been performed appropriately and rigorously? 

Reviewer #1: Yes

Reviewer #2: Yes

Reviewer #3: Yes

3. Have the authors made all data underlying the findings in their manuscript fully available?

Reviewer #1: Yes

Reviewer #2: Yes

Reviewer #3: Yes

4. Is the manuscript presented in an intelligible fashion and written in standard English?

Reviewer #1: Yes

Reviewer #2: Yes

Reviewer #3: No

5. Review Comments to the Author

Reviewer #1: General Comments:

The paper entitled ‘Increasing salinity of fibrinogen solvent generates stable fibrin hydrogels for cell delivery or tissue engineering’ describes a new method to generate fibrin hydrogels with desirable degradation properties through simple changes in the salinity of the precursor solution. The authors used a high salinity solution during the formation of the hydrogel to create an optically clear gel that has the ability to resist degradation and improved potential as a scaffold for the delivery of stem cells. The paper is well written and easy to read. The methods used to create the hydrogel are simple and have the potential to be used to generate tailored scaffolds. The paper suitable for publication if the following minor comments are addressed.

The paper needs to address the following minor concerns:

1. When referencing temperatures there needs to be consistency (sometimes the authors include the degree symbol and at other times not).

2. Figure 2C - Is there a statistical difference between your average fiber diameters for the PS and HS gels? While the SEM images look different, the data in the table doesn’t communicate if there is a significant difference.

3. Figure 3 – You should indicate where the live/dead stain and the stain for cell type change. You use the same color scheme and don’t indicate on the figure that they are different. This can lead to some confusion for the reader as the entire figure seems to be about the live/dead stain.

4. You don’t mention in your introduction or in the text why you chose to use papain as a degradation facilitator instead of plasmin. You need to explain your choice to use papain over plasmin when evaluating your degradation profiles?

5. You don’t provide gelation time data with increasing salinity. Gelation time data is fundamental and also particularly relevant when trying to understand fiber diameter and gel porosity data. If this data is available it should be included.

6. There is a spelling error in line 25 of the conclusions: “Howeer…”

7. Figure 5G – what does the tiny arrow indicate? There is nothing in the text or in the figure caption describing what the arrow is pointing to or why it is significant. This needs to be addressed.

Reviewer #2: Study goals were aimed at developing a slowly-degrading (“stable”) fibrin gel capable of supporting 1) the proliferation and differentiation of various stem cell types, 2) the development of a rudimentary capillary-like network in vitro, and 3) delivery and maintenance of cells in vivo without the need for degradation inhibitors. Studies generated fine, transparent high-salt (HS) fibrin gels and compared these with current opaque physiologic-salt (PS) fibrin gels. The HS fibrin gels degraded 2-3 36 times slower and did not interfere with cell viability. Overall, this is a very nice and well-organized study. Concerns are minor.

Major Comments:

It is unclear why these are called fibrin hydrogels (in the title and throughout) instead of PEG hydrogels since the PEG is the dominant concentration at 10:1.

Minor Comments:

1. For soft or viscous materials, typically a spherical AFM tip is recommended over a pyramidal tip.

2. Figure 1 – It is incorrect to present data from high concentrations to low concentrations. The low concentration should be closest to zero on the x-y axis.

3. You need to comment on why your results that fibrin opacity does not depend on CaCl2 concentration or pH were different compared with literature.

4. Fig 3 presents capillary-like network formations (Figs 3A-H) before viability. However, the viability is discussed first in the text. Please reverse the order in the figures so the viability data is first and aligns with the order discussed in the text.

5. If you want to increase impact, I suggest quantifying the vessel diameters in your gels. It appears that your HS gel is making finer sized vessels. The field has a hard time generating truly capillary-sized vessels, so this achievement could also be emphasized here.

6. It is surprising that you aren’t getting any vascular cell integration in vitro with either material. Please comment on why.

Reviewer #3: Title: Increasing salinity of fibrinogen solvent generates stable fibrin hydrogels for cell

delivery or tissue engineering.

Authors: Dillon K Jarrell, Ethan J Vanderslice, Mallory L Lennon, Anne C Lyons, Mitchell C VeDepo and Jeffrey Jacot

Summary: This research was done to improve the stability and lengthen the degradation period of fibrin gels.The contribution of this research lies in finding a way to enhance the stability of fibrin gels using a simple method of increasing the NaCl concentration and avoiding the use of fibrin degradation inhibitors, such as ε-aminocaproic acid, which have been shown to affect cellular processes.

General comments: The article does not read well and there were some inconsistencies that are further described below. There was an instance of overinterpretation and the flow of ideas in relation to figure presentation was not sequential which may confuse the reader. It is recommended that the authors carefully reevaluate the manuscript and consider the comments below.

Major comments:

• Swelling ratio section in methodology needs a little more explanation. Where the gels dehydrated first and then rehydrated? What was the formula used to calculate this ratio?

• Line 41 states that the reason of this homogeneity might be chlorine concentration that is able to alter the polymerization of 10mg/mL fibrinogen. This might be an overinterpretation since no other chlorine salt was used to measure the elastic moduli of these fibers. Furthermore, if chlorine is involved in the polymerization of fibrinogen, and if transparency and elastic modulus correlate, then increasing the calcium chloride concentration should have also increased transparency (figure 1B). However, that does not occur. What was the elastic moduli of the gels with variable calcium chloride concentrations?

• Line 42 page 16 the authors said, “to investigate this hypothesis, 200 mM gels were chosen for SEM analysis”. They should refer the reader to figures 2A and 2B. How does these images investigate the hypothesis that chloride ions are altering fibrinogen concentration? And why was 200 mM NaCl used instead of 250 mM?

• Figure 3 A to H are low quality images. Furthermore, it seems there is inconsistent magnification between the figures. Figure 3A seems to be lower magnification than figure 3B as the cells look smaller in A than in B. In the caption, the sentence “scale bars 50 um” is cut off (figure 3 caption page 29).

• Why were nude mice chosen for injection? Would it not be more suitable to use a healthy animal model to assess the immune response to such gels? The authors also mentioned that "the host innate immune system" (even though we are talking about nude mice) was able to degrade the PS gels. Are the authors suggesting that the HS gels evaded the host innate immune system? Would a healthy immune system degrade the HS gels as fast as the nude mice immune system degrades the PS gels?

• fig 6A-D. these figures do not clearly show spheroid formation. I suggest using a dotted line to outline the spheroids...

Minor comments:

• Minor spelling and grammatical mistakes found.

• Line 31 page 15 says 250 mM was chosen for the rest of the experiments yet in figure 2B and 2C 200 mM NaCl was chosen, even though in other figures 250mM was chosen again in other figures.

• Line 41 page 16 mentions that elastic moduli were more homogenous at NaCl concentrations greater than 200 mM. Assuming the authors are referring to the SD of the mean it seems that at 200 mM the fibers are also homogenous.

• Figures 4C and 4D are not mentioned in the text.

6. PLOS authors have the option to publish the peer review history of their article (what does this mean?). If published, this will include your full peer review and any attached files.

Reviewer #1: No

Reviewer #2: No

Reviewer #3: No

---

## [Author Response · Author response to Decision Letter 0]

17 Feb 2021

Reviewer #1: General Comments:

The paper entitled ‘Increasing salinity of fibrinogen solvent generates stable fibrin hydrogels for cell delivery or tissue engineering’ describes a new method to generate fibrin hydrogels with desirable degradation properties through simple changes in the salinity of the precursor solution. The authors used a high salinity solution during the formation of the hydrogel to create an optically clear gel that has the ability to resist degradation and improved potential as a scaffold for the delivery of stem cells. The paper is well written and easy to read. The methods used to create the hydrogel are simple and have the potential to be used to generate tailored scaffolds. The paper suitable for publication if the following minor comments are addressed.

The paper needs to address the following minor concerns:

1. When referencing temperatures there needs to be consistency (sometimes the authors include the degree symbol and at other times not).

We thank the reviewer for noticing this. This has been corrected.

2. Figure 2C - Is there a statistical difference between your average fiber diameters for the PS and HS gels? While the SEM images look different, the data in the table doesn’t communicate if there is a significant difference.

Thank you for catching this. There was indeed a significant difference in fiber diameter, and a depiction of this has been added to the table in Figure 2C.

3. Figure 3 – You should indicate where the live/dead stain and the stain for cell type change. You use the same color scheme and don’t indicate on the figure that they are different. This can lead to some confusion for the reader as the entire figure seems to be about the live/dead stain.

Thank you for this comment. We have added labels on the images in Figure 3 to help clarify what is being shown.

4. You don’t mention in your introduction or in the text why you chose to use papain as a degradation facilitator instead of plasmin. You need to explain your choice to use papain over plasmin when evaluating your degradation profiles?

Thank you for this comment. We agree that this should be discussed. We have now addressed this in Section 2.9: “Since we were not concerned with preservation of encapsulated proteins or cells, papain was selected as a protease for this experiment over the more expensive, fibrin-specific protease plasmin.”

5. You don’t provide gelation time data with increasing salinity. Gelation time data is fundamental and also particularly relevant when trying to understand fiber diameter and gel porosity data. If this data is available it should be included.

We thank the reviewer for this suggestion. We have performed the suggested experiment and included the results in Figure 2. We did not see a significant difference in gelation kinetics at 37 degrees C, which suggests that any ion-mediated inhibition of fibril aggregation does not slow the process of gelation/thrombin activity, but instead changes the way fibrils form. We have included the methods in a new Section 2.6, and have discussed the results in Section 3.2. 

6. There is a spelling error in line 25 of the conclusions: “Howeer…”

This has been corrected, thank you for noticing this.

7. Figure 5G – what does the tiny arrow indicate? There is nothing in the text or in the figure caption describing what the arrow is pointing to or why it is significant. This needs to be addressed.

We thank the reviewer for noticing this. We have added a description in the figure legend. It indicates a possible capillary forming from the implanted endothelial cells.

Reviewer #2: Study goals were aimed at developing a slowly-degrading (“stable”) fibrin gel capable of supporting 1) the proliferation and differentiation of various stem cell types, 2) the development of a rudimentary capillary-like network in vitro, and 3) delivery and maintenance of cells in vivo without the need for degradation inhibitors. Studies generated fine, transparent high-salt (HS) fibrin gels and compared these with current opaque physiologic-salt (PS) fibrin gels. The HS fibrin gels degraded 2-3 36 times slower and did not interfere with cell viability. Overall, this is a very nice and well-organized study. Concerns are minor.

Major Comments:

It is unclear why these are called fibrin hydrogels (in the title and throughout) instead of PEG hydrogels since the PEG is the dominant concentration at 10:1.

This is a good point and question. PEG is indeed the dominant molar concentration at 10:1, however is it the minority weight percent at 1:10. The amount of PEG added does not cause a gel to form, despite the fact that it does from crosslinks between fibrinogen proteins in solution. The gel only forms upon the addition of thrombin, demonstrating that the gel is predominantly fibrin. This has been clarified in the introduction. 

Minor Comments:

1. For soft or viscous materials, typically a spherical AFM tip is recommended over a pyramidal tip.

We thank the reviewer for this comment. We recognize that a spherical tip is recommended, however we used a tip that was readily available to us and cheaper than buying new spherical tips. We did not encounter any difficulties or abnormal behavior while using the pyramidal tip and so we did not feel purchasing spherical tips was necessary.

2. Figure 1 – It is incorrect to present data from high concentrations to low concentrations. The low concentration should be closest to zero on the x-y axis.

Thank you for noticing this. We have changed the graph in figure 1C to include a proper axis from low concentration to high.

3. You need to comment on why your results that fibrin opacity does not depend on CaCl2 concentration or pH were different compared with literature.

We thank the reviewer for this comment. We were exceedingly frustrated with our inability to recapitulate the results published in these other studies, especially reference #24. We have elaborated on possible causes for this in Section 3.1.

4. Fig 3 presents capillary-like network formations (Figs 3A-H) before viability. However, the viability is discussed first in the text. Please reverse the order in the figures so the viability data is first and aligns with the order discussed in the text.

Thank you for this comment. Figure 3 panels A-F indicate the Live/Dead results, and panels G and H show the network formation. Therefore, the Live/Dead results are discussed first in the text and appear first in the figure. We have added labels in Figure 3 panels A-H to clarify the difference between panels A-F and G-H.

5. If you want to increase impact, I suggest quantifying the vessel diameters in your gels. It appears that your HS gel is making finer sized vessels. The field has a hard time generating truly capillary-sized vessels, so this achievement could also be emphasized here.

This is an interesting point, thank you for this comment. We may investigate this in future work using this hydrogel. However, we did not find any evidence demonstrating that the “capillary-like networks” are actually vessels with lumen, but rather are string-like networks of endothelial and stromal cells. Because we do not think any lumen are present, we prefer not to quantify diameters. We did, however, add this comment as a discussion point at the end of the second paragraph of section 3.3.

6. It is surprising that you aren’t getting any vascular cell integration in vitro with either material. Please comment on why.

This is a great point. We have commented on why this may be at the end of section 3.4.

Reviewer #3: Title: Increasing salinity of fibrinogen solvent generates stable fibrin hydrogels for cell

delivery or tissue engineering.

Authors: Dillon K Jarrell, Ethan J Vanderslice, Mallory L Lennon, Anne C Lyons, Mitchell C VeDepo and Jeffrey Jacot

Summary: This research was done to improve the stability and lengthen the degradation period of fibrin gels. The contribution of this research lies in finding a way to enhance the stability of fibrin gels using a simple method of increasing the NaCl concentration and avoiding the use of fibrin degradation inhibitors, such as ε-aminocaproic acid, which have been shown to affect cellular processes.

General comments: The article does not read well and there were some inconsistencies that are further described below. There was an instance of overinterpretation and the flow of ideas in relation to figure presentation was not sequential which may confuse the reader. It is recommended that the authors carefully reevaluate the manuscript and consider the comments below.

Major comments:

• Swelling ratio section in methodology needs a little more explanation. Where the gels dehydrated first and then rehydrated? What was the formula used to calculate this ratio?

Thank you for this comment. We have indeed elaborated this methodology in section 2.5.

• Line 41 states that the reason of this homogeneity might be chlorine concentration that is able to alter the polymerization of 10mg/mL fibrinogen. This might be an overinterpretation since no other chlorine salt was used to measure the elastic moduli of these fibers. Furthermore, if chlorine is involved in the polymerization of fibrinogen, and if transparency and elastic modulus correlate, then increasing the calcium chloride concentration should have also increased transparency (figure 1B). However, that does not occur. What was the elastic moduli of the gels with variable calcium chloride concentrations?

This is an insightful comment. As we thought more about this, we realized that in Figure 1B we only increased CaCl2 to 25 mM, corresponding to a total final chloride concentration of 195 mM when using 145 mM NaCl. Therefore, we decided to do additional experiments to further increase the CaCl2 used in gel fabrication to see if we could recapitulate the transparency seen when we used high NaCl concentrations. Interestingly, increasing CaCl2 without increasing NaCl did not increase the transparency of the gels, even when the total Cl- level from CaCl2 addition was greater than the Cl- level of the transparent HS gels achieved using NaCl addition (270 mM Cl- in HS gels using increased NaCl; 275 mM Cl- in newly-fabricated gels using increased CaCl2). We thought that this may be due to the white precipitate formed by the reaction of the calcium and phosphate groups, but even after centrifuging the solutions prior to gelation in order to eliminate the precipitate the gels remained opaque. The light absorbances of these additional gel formulations have been added to Figure 2. We do not think it is necessary to assess the elastic moduli of these additional formulations because the gels remained opaque. We have added additional discussion for these points in Section 3.1. 

• Line 42 page 16 the authors said, “to investigate this hypothesis, 200 mM gels were chosen for SEM analysis”. They should refer the reader to figures 2A and 2B. How does these images investigate the hypothesis that chloride ions are altering fibrinogen concentration? And why was 200 mM NaCl used instead of 250 mM?

We have taken new SEM images of the 250 mM NaCl gels because we agree that this fits better with the paper. Our hypothesis that chlorine ions are altering the fibrin morphology was not the primary hypothesis of the paper, but was instead a possible explanation for the differences in morphology that we observed. Therefore, we have changed Figure 2B to the 250 mM SEM image and removed the sentence you cited for clarity.

• Figure 3 A to H are low quality images. Furthermore, it seems there is inconsistent magnification between the figures. Figure 3A seems to be lower magnification than figure 3B as the cells look smaller in A than in B. In the caption, the sentence “scale bars 50 um” is cut off (figure 3 caption page 29).

As whole-gel images, Figure 3 A-H are the clearest images we could achieve at that magnification. Magnifications of the panels were double-checked and corrected to ensure all images were of equal magnification. Any remaining differences in apparent cell size are due to the fact that the cells are distributed in 3D throughout the gel, and therefore some may appear out of focus and larger or smaller.

• Why were nude mice chosen for injection? Would it not be more suitable to use a healthy animal model to assess the immune response to such gels? The authors also mentioned that "the host innate immune system" (even though we are talking about nude mice) was able to degrade the PS gels. Are the authors suggesting that the HS gels evaded the host innate immune system? Would a healthy immune system degrade the HS gels as fast as the nude mice immune system degrades the PS gels?

These nude mice were athymic, meaning that they retain their innate immune system but do not have an adaptive immune system. We chose this animal model because we were implanting human cells, which would be promptly killed by an adaptive immune system but not an innate immune system. We are not suggesting that the HS gels evaded the innate immune system; rather we believe that both the HS and PS gels were exposed to the same macrophages and proteases but that the HS gels were less affected, as indicated by their longevity in vivo. We would not expect the presence of an adaptive immune system to affect the fibrin degradation rate, but we would expect it to kill any foreign cells.

• fig 6A-D. these figures do not clearly show spheroid formation. I suggest using a dotted line to outline the spheroids...

Thank you for this comment. In section 3.5 and the Figure 6 legend we changed the wording so as not to indicated actual spheroids. Rather, we observed compact 3D colonies in the HS gels and not in the PS gels.

Minor comments:

• Line 31 page 15 says 250 mM was chosen for the rest of the experiments yet in figure 2B and 2C 200 mM NaCl was chosen, even though in other figures 250mM was chosen again in other figures.

This has been corrected, and the figure in 2B was replaced with the 250 mM image.

• Line 41 page 16 mentions that elastic moduli were more homogenous at NaCl concentrations greater than 200 mM. Assuming the authors are referring to the SD of the mean it seems that at 200 mM the fibers are also homogenous.

This is a good catch. We have corrected this sentence to “greater than or equal to 200 mM”.

• Figures 4C and 4D are not mentioned in the text

Thank you for this catch. This has been corrected in Section 3.3. We also noticed a few other panels that were not mentioned in the text and corrected this as well.

We hope that the edits made to the manuscript and explanations given here are sufficient to progress this work towards publication in PLoSONE. Thank you all again for your time and thoroughness.

---

## [Decision Letter · Decision Letter 1]

15 Mar 2021

Increasing salinity of fibrinogen solvent generates stable fibrin hydrogels for cell delivery or tissue engineering

PONE-D-20-26975R1

Dear Dr. Jacot,

We’re pleased to inform you that your manuscript has been judged scientifically suitable for publication and will be formally accepted for publication once it meets all outstanding technical requirements.

Kind regards,

Esmaiel Jabbari, PhD

Academic Editor

PLOS ONE

Additional Editor Comments (optional):

Reviewers' comments:

Reviewer's Responses to Questions

**Comments to the Author**

1. If the authors have adequately addressed your comments raised in a previous round of review and you feel that this manuscript is now acceptable for publication, you may indicate that here to bypass the “Comments to the Author” section, enter your conflict of interest statement in the “Confidential to Editor” section, and submit your "Accept" recommendation.

Reviewer #1: All comments have been addressed

Reviewer #3: All comments have been addressed

2. Is the manuscript technically sound, and do the data support the conclusions?

Reviewer #1: Yes

Reviewer #3: Yes

3. Has the statistical analysis been performed appropriately and rigorously? 

Reviewer #1: Yes

Reviewer #3: Yes

4. Have the authors made all data underlying the findings in their manuscript fully available?

Reviewer #1: Yes

Reviewer #3: Yes

5. Is the manuscript presented in an intelligible fashion and written in standard English?

Reviewer #1: Yes

Reviewer #3: Yes

6. Review Comments to the Author

Reviewer #1: It is the opinion of this reviewer that all comments have been adequately addressed and the article is acceptable for publication.

Reviewer #3: The authors have addressed my previous comments. The manuscript is improved and I recommend accepting it.

7. PLOS authors have the option to publish the peer review history of their article (what does this mean?). If published, this will include your full peer review and any attached files.

Reviewer #1: No

Reviewer #3: No

---

## [Editor Report · Acceptance letter]

7 May 2021

PONE-D-20-26975R1 

Increasing salinity of fibrinogen solvent generates stable fibrin hydrogels for cell delivery or tissue engineering 

Dear Dr. Jacot:

I'm pleased to inform you that your manuscript has been deemed suitable for publication in PLOS ONE. Congratulations! Your manuscript is now with our production department. 

Kind regards, 

on behalf of

Dr. Esmaiel Jabbari 

Academic Editor

PLOS ONE